# Sedimentary Mn Metallogenesis and Coupling among Major Geo-Environmental Events during the Sturtian Glacial–Interglacial Transition

**Liping Liu [1], Zuzhou Jiang [2] and Fengyou Chu [1],***

[1] Key Laboratory of Submarine Geosciences, Second Institute of Oceanography, Ministry of Natural Resources, Hangzhou 310012, China

[2] Construction Administration Bureau of Water Diversion Irrigation Area of Drought Harnessing Letan Reservoir in Central Guangxi, Laibin 546100, China

\* Correspondence: chu@sio.org.cn

**Abstract:** The Sturtian (720–670 Ma) glacial–interglacial transition period was an important interval for sedimentary manganese metallogenesis, including the Mn oxide deposit in the Otjosondu region in Namibia and Mn carbonate deposits in the Datangpo Formation in the south-eastern Yangtze Platform, South China. During this period, Earth experienced the breakup of Rodinia, the Sturtian glaciation, and the Neoproterozoic oxygenation event. In this study, we investigate scenarios that might have provided geologically and geochemically favorable conditions for Mn metallogenesis. In these scenarios, the global recovery of microorganisms enhanced marine primary productivity and $O_2$ levels of the hydrosphere and atmosphere during the Sturtian glacial–interglacial transition. However, the water column was not completely oxidized, maintaining redox stratification. Transgression–regression cycles or $O_2$-rich downwelling drove the exchange of oxygenated topwater and anoxic deep water in rift-related basins that developed due to Rodinia's breakup. The coupling of these processes precipitated existing dissolved Mn(II) at the margins of basins (Otjosondu region) or at their centers (Yangtze Platform). In the latter case, precursor Mn oxides were further converted into Mn carbonates via the reduction of Mn oxides coupled with organic matter oxidation during early diagenesis. A brief review of Mn metallogenesis in the geological record reveals that Mn metallogenic processes typically occur under geo-environmental conditions that, in concert, produce favorable conditions for Mn sourcing, concentration, and sedimentation.

**Keywords:** Sturtian glaciation; Mn metallogenesis; Datangpo Formation; Chuos Formation; early diagenesis

## 1. Introduction

Manganese metallogenesis in sedimentary rocks over the course of Earth's evolution has been primarily controlled by redox changes [1,2]. Similar to iron, Mn is a transition element that undergoes active biogeochemical cycles. Under anoxic conditions and a low pH (<5) in aquatic systems, dissolved Mn(II) is the most stable and dominant Mn species, whereas under highly oxic conditions, insoluble oxidized Mn species (Mn(III) and Mn(IV)) are thermodynamically favored [3,4]. Therefore, Mn(II) converts to solid-phase Mn oxides or hydroxides when it encounters oxidizing conditions or crosses the redox interface in seawater between oxic and dysoxic water masses.

The breakup of the Rodinian supercontinent, the Sturtian glaciation, and the Neoproterozoic oxygenation event (NOE) occurred during the Cryogenian period (720–635 Ma) in the mid-Neoproterozoic era [3,5–7]. In these scenarios, interstratified Mn oxides and banded iron formations were formed in the Damara sequence in the Otjosondu region during the Sturtian glacial–interglacial transition [2,3], as were black shale–hosted Mn

carbonate deposits in the basal Datangpo Formation in the south-eastern Yangtze Platform, China [4,6,8,9].

In this study, we investigate these major geo-environmental events and their coupling effects in these two sedimentary Mn deposit formations during the Sturtian glacial–interglacial transition. We also compile a series of published data, including major and rare-earth elements and stable carbon and oxygen isotopes of Mn ore and host rock samples from the Datangpo Formation and selected Mn deposits throughout the geological record. Based on this analysis, we discuss the metallogenic mechanisms relevant to Sturtian Mn deposits. Finally, we perform a brief review of major sedimentary Mn deposits throughout Earth's history and present general models for large-scale sedimentary Mn enrichment.

## 2. Initiation and Termination of Sturtian Glaciation

In the context of the breakup of Rodinia, Neoproterozoic Earth experienced several glacial events. The most severe and long-lasting glacial events were the Sturtian (720–670 Ma) and Marinoan ones (650–635 Ma) [10–12]. The fundamental factor that is believed to be responsible for extreme Sturtian glaciation is the spectacular tectonic–magmatic evolution that occurred in response to tectonic rifting during the breakup of Rodinia [13]. The weathering of the basaltic provinces, especially those located within the equatorial region, drastically induced atmospheric $CO_2$ drawdown and prompted a global transition into a cooling stage [14]. Moreover, the increased availability of moisture along the continental borders of new smaller plates led to increased terrestrial runoff, thereby boosting global silicate weathering, and thus, further decreasing the level of global atmospheric $CO_2$ [14,15].

However, mantle-plume-related volcanism continued during the Sturtian period [16]. $CO_2$ slowly accumulated in the atmosphere until the ensuing greenhouse effect was capable of triggering the transition to the Sturtian interglacial period [11]. Furthermore, the southward drift of the large Laurentian magmatic province away from the equatorial region, where magmatic rock weathering and $CO_2$ consumption had been boosted by optimal climatic conditions, mitigated continental-weathering-related $CO_2$ consumption, thus preventing an endless succession of cyclic snowball glaciations [11,13,17].

The ice cover acted as a barrier to the exchange of mass and oxygen between the ocean and the atmosphere [11,14]. Ongoing rift-related volcanism and hydrothermal activity during the Sturtian glaciation, or even earlier, introduced massive amounts of dissolved Mn(II) into the anoxic water of the rift basins, but its precipitation was prevented by global conditions prior to the Mn metallogenesis era after Sturtian glaciation [16].

The global synchroneity of the onset of Sturtian glaciation is evident from geochronological data collected from several cratons of the Rodinia supercontinent, including Laurentia ($717 \pm 4$ Ma, Pocatello Formation) [18], Arabia ($713.7 \pm 0.5$ Ma, Ghubrah Formation) [19], and South China (717–714 Ma) [20–22]. Sturtian glacial diamictites at many locales are sharply overlain by post-Sturtian cap carbonates (Figure 1). Many geochronological investigations of the interglacial black shales or clastic rocks near the Sturtian diamictites or Sturtian cap carbonates (Figure 1a), including Re–Os isotope and U–Pb-sensitive high-resolution ion microprobe (SHRIMP) zircon dating studies, suggest a globally synchronous end to the Sturtian glaciation (657–667 Ma).

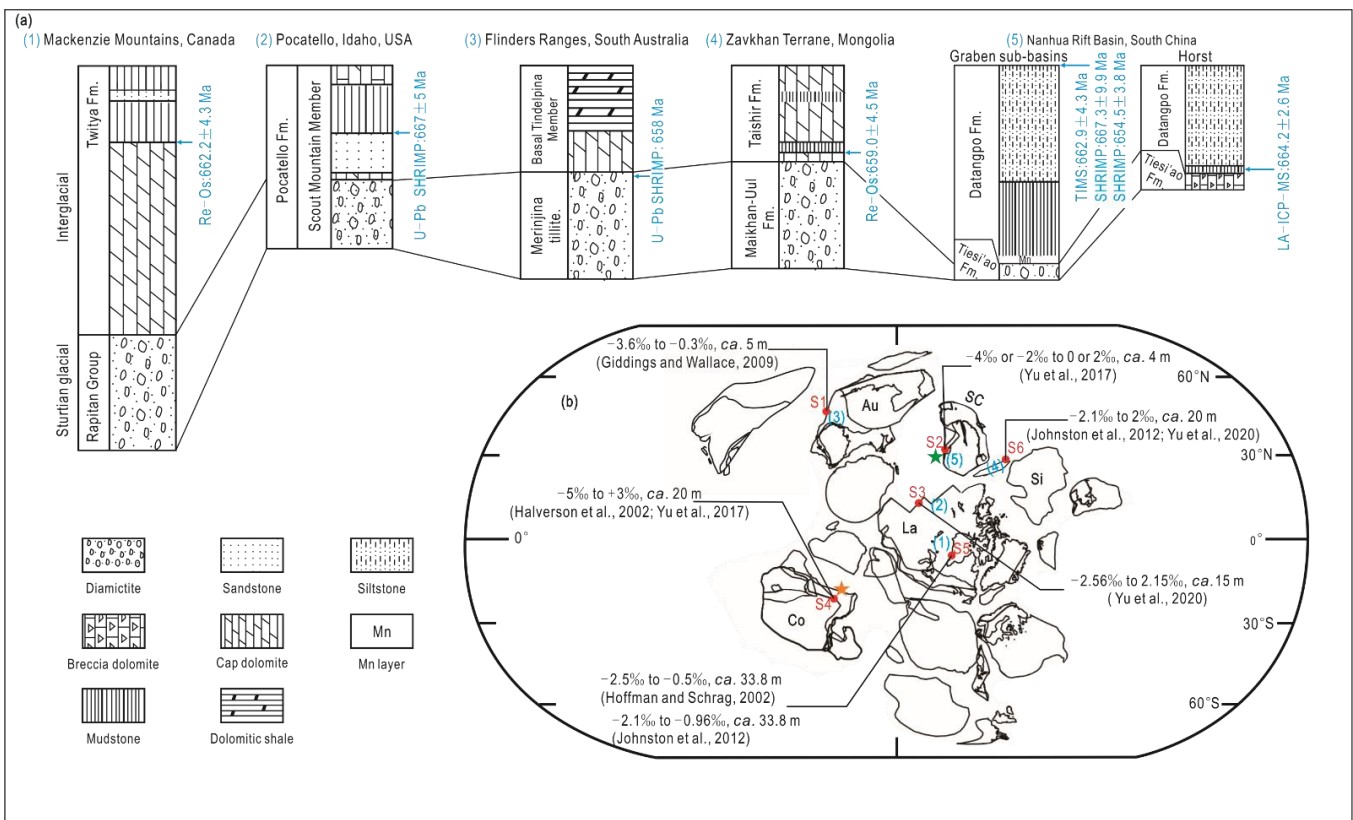

**Figure 1.** (**a**) Schematic stratigraphy and the end of Sturtian glaciation worldwide (modified from [23,24]). Columns (1) and (3) were adapted from [25–27]; column (2) was adapted from [18]; column (4) was adapted from [28,29]. (**b**) Paleogeographic reconstruction of Rodinia at approximately 680 Ma, with $\delta^{13}C_{carb}$ values from the bottom to the top of post-Sturtian cap carbonates as distributed across cratons. Post-Sturtian cap carbonate locations are shown as S1–S6. The green star represents the Mn deposit location in the Datangpo Formation in the south-eastern Yangtze Platform, South China, while the orange star represents the Mn deposit location in the Otjosondu region in Namibia. Craton abbreviations: Au–Australia; Co–Congo; La–Laurentia; Si–Siberia; SC–South China.

## 3. Geo-Environmental Events during the Sturtian Glacial–Interglacial Transition

In the context of the breakup of Rodinia and the Sturtian glacial–interglacial transition, dramatic exogenic environmental changes occurred across the Earth [5,12,30]. The coupling of these geological and environmental processes powerfully influenced the evolution of atmospheric oxygen and new environments for early life [31]. This same coupling also provided favorable conditions for Mn transportation, accumulation, and enrichment.

### 3.1. Microorganismal Recovery

The Sturtian glaciation may not have caused evolutionary bottlenecks, as assumed previously. For example, phytane and pristane as biomarkers of eukaryotes have been found in Sturtian tillites in the Yangtze region, though at very low concentrations, suggesting at least minimal photosynthesis occurred during the Sturtian glaciation [32]. Thus, it has been assumed that cryoconite holes (Figure 2a) and crack systems incorporated sea ice in their channels. These channels in the modern polar ocean are known to have been inhabited by eukaryotes, protists, and prokaryotes and may have served as reliable refugia for early microorganisms [12,33]. Figure 2a shows a model in a cryoconite pond as a habitat for eukaryotes. Cryoconite and meltwater would have flushed through moulins within the ice cover. Flushed cryoconite organic matter would have been subjected to anaerobic respiration coupled with sulfate or Fe(III) reduction in the water columns and

sediments. The organic matter would have been buried, while $O_2$ would have been added to the atmosphere.

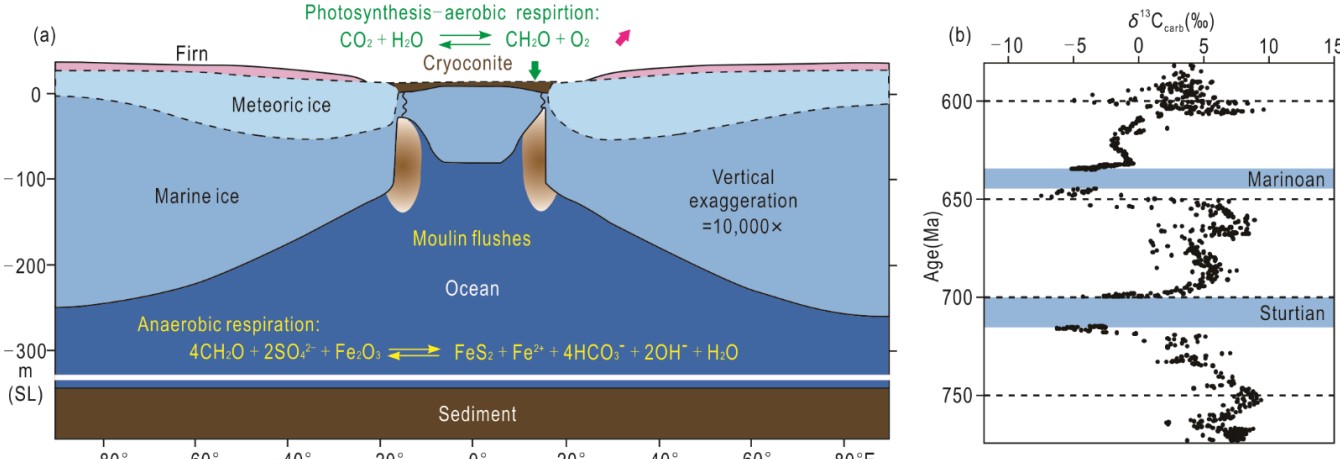

**Figure 2.** (**a**) Schematic diagram of a cryoconite hole. (**b**) Variation in the $\delta^{13}C_{carb}$ values of marine carbonates from 750 to 600 Ma. ((**a**) was taken from [12]; (**b**) was modified from [5,34,35]). SL: sea level.

Spectacular, far-reaching biological events may have occurred immediately after the Sturtian glaciation. Massive nutrient flows (e.g., P and Fe) derived from intensive ongoing submarine volcanism [16] and the erosion of supermountains created by continental collisions during the breakup of Rodinia [36] might have significantly influenced the oceans' chemistry and triggered a drastic increase in primary productivity.

Most post-Sturtian cap carbonates contain microbially mediated structures and microbial fossils, such as microbial mats, tube-like structures, and/or roll-up structures, which are regarded as evidence for the recovery of microbial communities in the aftermath of the Sturtian glaciation [29,37,38]. In this study, we reviewed $\delta^{13}C_{carb}$ values of the postglacial cap carbonates globally overlying glacigenic sequences of the Sturtian period (Figure 1b). The results show that these cap carbonates yield negative $\delta^{13}C$ values (−5‰ to −2‰) at their bases, increasing to weakly positive values (0 to +3‰) at their tops (Figure 1b). This pattern suggests the recovery of marine primary productivity during the interglacial interval [29].

### 3.2. Atmospheric Oxygen Enhancement

Oxygenic photosynthesis by algae and cyanobacteria significantly increased the $O_2$ levels [36]. Moreover, high sedimentation rates have been suggested to promote the rapid burial of organic matter that emerged due to high productivity. These factors, along with the introduction of less easily degradable organic matter and the expansion of eukaryotic life, prevented large quantities of organic matter from reacting with oxygen [39,40]. The occurrence of positive $\delta^{13}C_{carbonate}$ excursion immediately after the Sturtian glaciation (Figure 2b) implies the enhanced burial of organic carbon and, therefore, a rise in atmospheric oxygen levels. This enhanced oxygen level provided an opportunity for the oxygenation of hydrosphere and subsequent Mn precipitation and enrichment.

### 3.3. Redox-Stratified Water System

Although the level of $pO_2$ on the Earth's surface during the waning of Sturtian glaciation was probably higher than that of 1% PAL (present atmospheric level) [41], the seawater system was not completely oxidized and exhibited redox stratification, with oxygen-rich topwater and anoxic deep water [42]. Moreover, the bottom-to-top increase in the $\delta^{13}C$ values of post-Sturtian cap carbonates may be another signature of the redox-stratified state of the ocean system (Figure 1b).

For example, many studies of the redox state of seawater during this period have focused on the Nanhua Rift Basin in the south-eastern Yangtze Platform, where Mn deposits in the Datangpo Formation exist [43–50]. A series of proxies used to reconstruct the redox environment of the Datangpo Formation indicate that the deep water column in the Nanhua Rift Basin might have experienced a redox transition from an anoxic state during the Sturtian glaciation to a temporarily oxygenated state during the Mn metallogenic era (Figure 3). Mn enrichment in the basal Datangpo Formation is thus an indicator of oxic conditions.

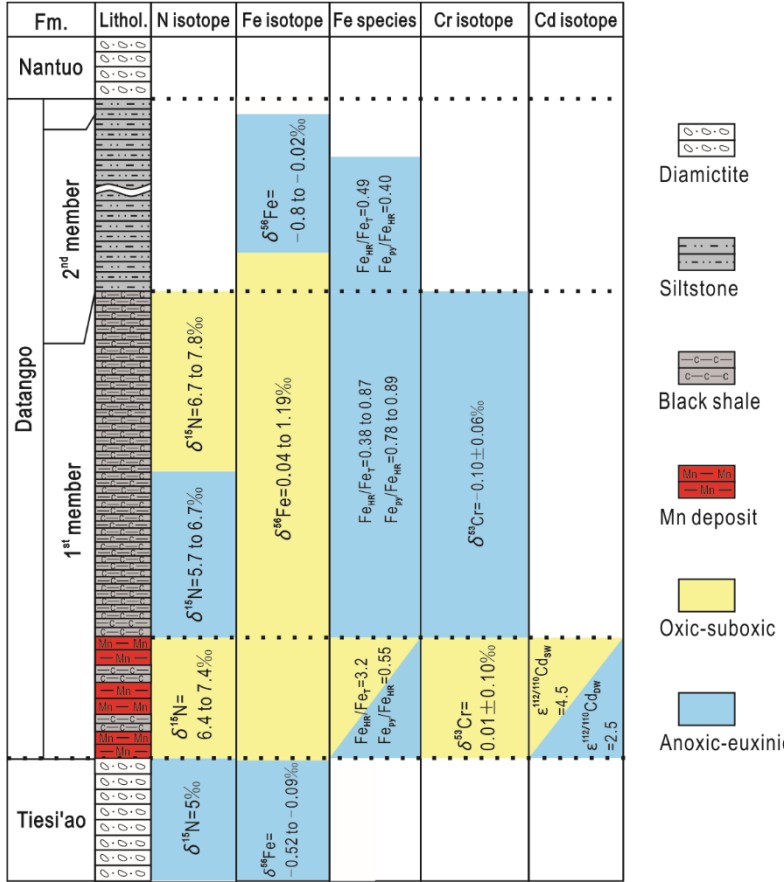

**Figure 3.** Elemental and isotopic proxies for reconstructing paleoenvironmental oxygenation during deposition of the Sturtian-age Tiesi'ao Formation and the 1st (Mn ores and black shales overlying the Mn layers) and 2nd members of the Datangpo Formation (data sources: N isotopes [44]; Fe species [46]; Fe isotopes [48]; Cr isotopes [49]; Cd isotopes (SW: surface water; DW: deep water) [50]).

These episodic water redox shifts between anoxic and oxic conditions on the seafloor or in deep basins at other locales might not have been rare during the Sturtian interglacial interval. For example, extensive but temporary seafloor oxygenation is evidenced by the high $\delta^{238}$U values of post-Sturtian limestones in the Taishir Formation, Mongolia [43]. Oxygen might have been consumed via the subsequent oxidation of organic matter, returning the ocean to an anoxic state. Coincidentally, if this redox transition happened to encounter substantial amounts of dissolved Mn(II) stored in the anoxic water column within the relatively enclosed basins, episodic ventilation with cold oxygenated surface water would then have induced large-scale Mn sedimentation.

## 4. Mn Metallogenesis during the Sturtian Glacial–Interglacial Transition

### 4.1. Mn Deposits

#### 4.1.1. Mn Deposit in the Otjosondu Region

The Otjosondu region in Namibia hosts the only known occurrence of economic Mn deposits, which correlate with the widespread banded iron formations (BIFs) that exist within the Chuos Formation. The Chuos Formation is found in the middle of the Late Proterozoic Damara sequence, which is composed mostly of shelf glaciomarine sediments [3,51]. From bottom to top, the sequence of the Chuos Formation consists of lower supermature quartz arenites, a lower Mn horizon, BIFs, an upper Mn horizon, and upper supermature quartz arenites (Figure 4a). The lower and upper Mn ore horizons were formed during transgressive and regressive stages, respectively, in association with glacial–interglacial cycles [2,3]. Hydrothermal activity, linked to the initial spreading of the Khomas Sea, represents the most likely candidate for the Mn source [3]. During transgressions and regressions, remanent dissolved metals (Mn(II) and Fe(II)) in the pH–buffered seawater encountered the redox interface developing between the basinal waters and oxygenated shelf waters in the outer shelf area (Figure 4b). The cyclic depositional rhythm of BIFs and Mn layers, resembling sandwiches, was a result of the shifting of Fe(II)/Fe(III) and Mn(II)/Mn(IV) redoxclines as a result of repeated transgression–regression cycles. Mn would have been precipitated in more highly oxidizing, shallower shelf environments than those of Fe. The sandwiched ore-bearing sequences thus likely represent a complete large-scale transgressive–regressive cycle. Mn-rich deposits in the upper and lower horizons exist in lithofacies-type sequences whose orders are mutually reverse (Figure 4b). Moreover, small-scale sedimentary cycles can be interpreted as glacial-induced sea-level fluctuations that changed the location of the redox boundary along the shelf.

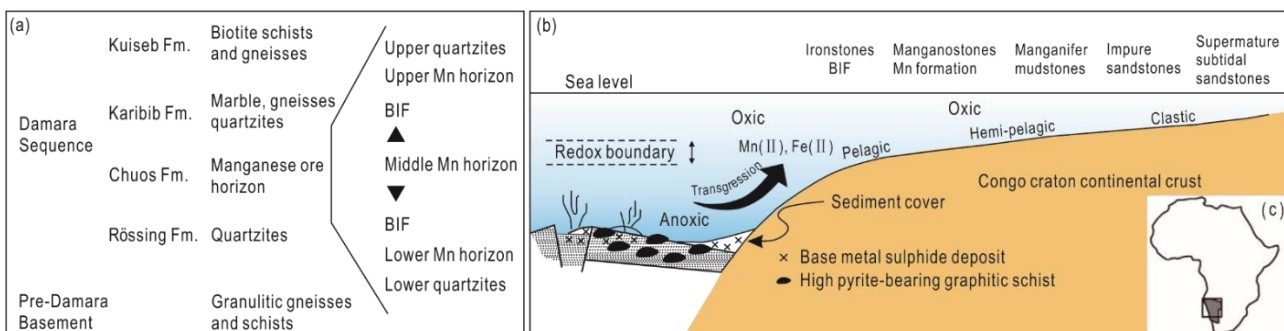

**Figure 4.** (**a**) Stratigraphic overview of the Damara Sequence, Namibia (modified from [2,3]). (**b**) Schematic diagram of the formation of interbedded Mn oxides and banded iron formations in the Damara sequence in the Otjosondu region in Namibia. Ore-forming solutions were likely derived from hydrothermal vents, which developed in the newly formed oceanic crust in the Khomas trough to the south of the Otjosondu shelf (modified from [3]). (**c**) Location of Namibia in Africa.

#### 4.1.2. Mn Deposits in the South-Eastern Yangtze Platform

Mn-rich sequences interbedded with black carbonaceous shales are hosted in the first member of the Datangpo Formation in a series of graben sub-basins in the Nanhua Rift Basin in the south-eastern Yangtze Platform (Figure 5a). Mn carbonate ore deposits are restricted to the deepest parts of a series of graben sub-basins of the Nanhua Rift Basin, rather than around the sub-basin margins [6–9]. The detailed locations of these Mn deposits are shown in Figure S1 in Supplementary File S1.

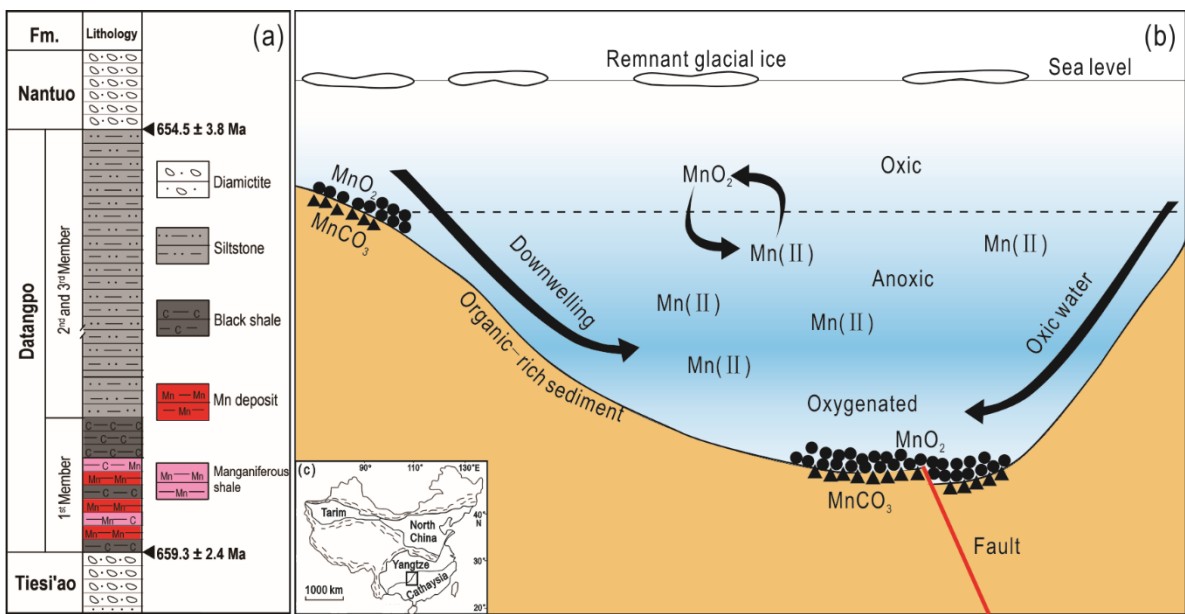

**Figure 5.** (**a**) Stratigraphic section showing the Mn beds in the basal Datangpo Formation (modified from [23,52]). (**b**) "Episodic ventilation" model for the formation of Mn carbonate deposits in the sub-basins of the Nanhua Rift Basin, South China (adapted from [6]). (**c**) Location of the Mn deposit in China.

We compiled published data on major rare-earth elements (REEs + Y and REY) in Mn carbonate ores and black carbonaceous shale host rocks from the first member of the Datangpo Formation (Supplementary File S2—original data and references). The post-Archean Australian Shale (PAAS)-normalized REY pattern (Figure 6a) shows that the Mn ores have an obvious enrichment of middle REEs, resulting in a "hat-shaped" plot, whereas black shales (Figure 6b) have slightly higher more REE enrichment, resulting in a flat plot. Mn ore samples are characterized by prominent positive Ce anomalies, with $Ce_{SN}/Ce_{SN}^{*}$ values ranging from 0.99 to 1.38 (average: 1.22). These findings indicate an Mn oxide precursor for Mn carbonate ores. Meanwhile, the lack of any obvious Ce anomaly in host rocks (average: 1.07; Supplementary File S2) implies an anoxic depositional environment. Other redox proxies (e.g., U, Mo, and V) analyzed in the entire profile of the Datangpo Formation [6,23,53] also suggest that Mn-enriched layers were deposited in oxic environments, whereas interbedded black carbonaceous shale host rocks were deposited in anoxic conditions. Moreover, positive Eu anomalies (Figure 6a; average: 1.16) and extremely low Al/Mn values of Mn layers (0.02–0.57) (Supplementary File S2) indicate that the original Mn source was most likely derived from hydrothermal fluids.

Mn accumulation in the deepest parts of graben sub-basins may be due to the intermittent intrusion of cold oxygenated surface water in the redox-stratified Nanhua Rift Basin during the Sturtian glacial–interglacial transition. This would seem to be the only mechanism capable of forming a sedimentary Mn deposit in the center of the ore-hosting basin. This situation is similar to that in the modern Baltic Sea [6,54,55], where episodic inflows of dense, oxygenated water masses from the North Sea intrude at intervals of several years, and Mn(II) is oxidized and preferentially accumulates during stagnation periods in the deepest basins, where currents have no influence. However, the driving factor for the redox (oxic–anoxic) exchange of the stratified water column in the Nanhua Rift Basin may be the possible presence of episodic cold events interrupting the warming process during the Sturtian glacial–interglacial transition period [6,55]. Sea ice might have remained during the aftermath of the Sturtian glaciation, and complete deglaciation might not have been as abrupt as previously thought it was. Thus, additional sea ice formation, which makes surface water denser due to brine rejection, would contribute to replenishing

anoxic bottom water (Figure 5b). This possibility is supported by the following line of evidence. (1) A plot depicting chemical index of alteration (CIA) data for the first member of the Datangpo Formation displays a serrated, upward trend (e.g., Figure 7 in [52]). This trend might reflect a tortuous process toward achieving a relatively stable warm climate after the Sturtian glaciation, given that the CIA can serve as a proxy in climate change reconstruction [52,55,56]. (2) At some locations, the basal Datangpo Formation contains at least two diamictite layers, each 20 cm thick [23]. Moreover, the presence of dropstone-bearing intervals just above the base of the Sturtian-aged Tindelpina cap carbonate in South Australia indicates a similar situation [27]. The multiplicity of these layers should be understood as resulting from recurring glacigenic debris flows caused by multiple ice-melting events rather than a single sudden melting event.

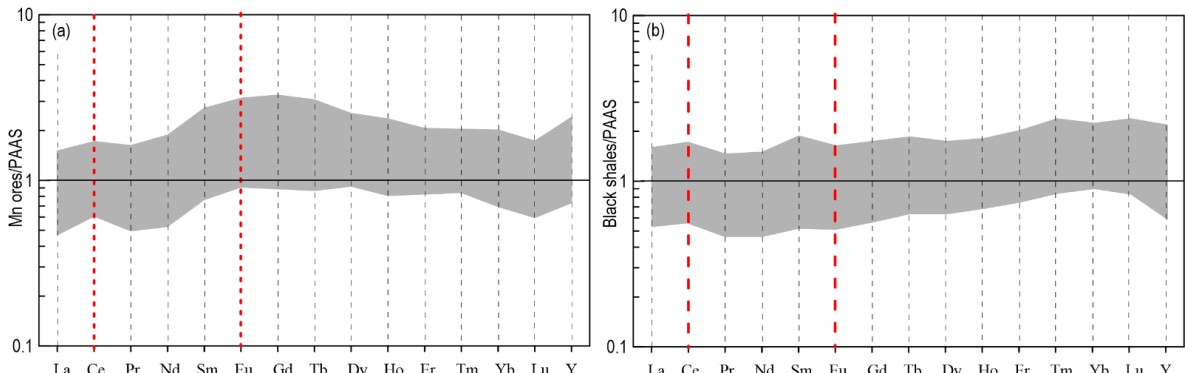

**Figure 6.** REY patterns normalized to PAAS for (**a**) Mn ores and (**b**) host rocks (black shales) of the first member of the Datangpo Formation. Gray boundaries are based on the distributions of data presented in Supplementary File S2.

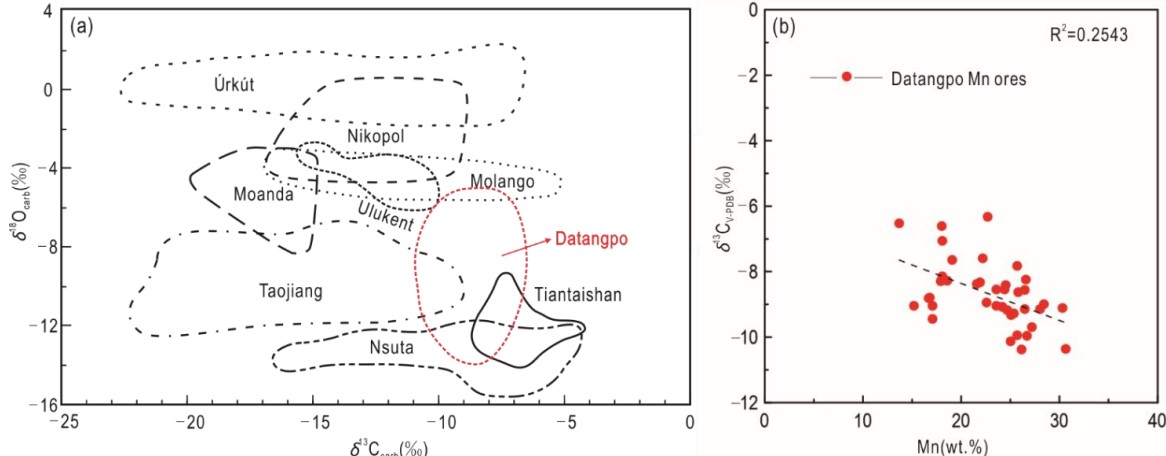

**Figure 7.** (**a**) Plot of carbon and oxygen isotope values for major Mn carbonate deposits in the geologic record (modified from [9,57]; Mn carbonate data for the following deposits obtained from the following sources: Nikopol [57]; Ulukent [58]; Úrkút [59]; Molango [60]; Taojiang [61]; Moanda [57]; Tiantaishan [57]; Nsuta [62]). (**b**) Scatterplot of Mn content versus $\delta^{13}C_{carb}$ value for Mn carbonate ores of the Datangpo Formation in the south-eastern Yangtze Platform in South China (data obtained from [6,9,63]).

Notably, Mn carbonate rather than Mn oxide characterizes the Mn deposits in the Datangpo Formation. Negative $\delta^{13}C_{carb}$ values (Figure 7) and the negative relationship between Mn contents and $\delta^{13}C_{carb}$ values (Figure 7b) imply that precursor Mn oxides were reduced by organic matter, whose $\delta^{13}C_{org}$ values typically range from −34‰ to −31‰. This process would have released Mn(II) and bicarbonate ($HCO_3^-$), which would have

served as the precursors for Mn carbonate precipitation. Mn carbonate precipitation in the basal Datangpo Formation might have occurred in an early diagenetic pore-water environment where considerable bicarbonate was sourced from seawater [9]. This possibility is supported by the very low $\delta^{13}C_{carb}$ of Datangpo Mn carbonate ores in comparison with some other Mn carbonate ore deposits in the geologic record, each of which exhibits a broad range of $\delta^{13}C_{carb}$ values (Figure 7a).

### 4.2. Contribution of Microbial Processes to Mn Metallogenesis

In addition to the geochemical behavior of Mn, microbially mediated Mn fixation plays an important role in considerable Mn accumulation in sediments. Mn deposits across different geological periods exhibit similar two-step microbial metallogenesis processes [4,64]. (1) The primary chemolithoautotrophic cycle: microbes oxidize Mn(II) to Mn(IV) via an enzymatic pathway under essential oxic conditions. (2) The diagenetic heterotrophic cycle: this cycle involves the decomposition and mineralization of cells and extracellular polymeric substances (EPSs) of Mn (and Fe) bacteria.

In the case of Mn carbonate deposits within black shale, microbes mediate Mn-oxide reduction via organic matter decomposition, resulting in the formation of rhodochrosite, characterized by a low $\delta^{13}C_{carb}$ signal, as the final product [7,8]. However, for Mn-oxide deposits, such as those found in the Otjosondu region of Namibia, Mn oxides do not transform into Mn carbonates, either due to them having a slow organic matter sedimentation rate or because the majority of the organic matter was oxidized under oxic conditions throughout the process of metallogenesis.

Although diagenesis and other processes can obscure microbial characteristics, remnants of microbial activity, including mineralized microbial microtextures and fossils of various shapes [4,7,64], were preserved, likely owing to the protective nature of the minerals around the cells and EPS. Polgári and Gyollai [8] suggest using a multiscale methodology to verify the role of microbe-mediated ore-forming processes.

### 4.3. Mn and Fe Fractionation

The fractionation of Mn with respect to Fe is a prerequisite for forming sedimentary Mn deposits [64]. Fe(II)/Fe(III) redoxcline is achieved at much lower $E_H$ and pH levels than that which is required for the Mn(II)/Mn(IV) buffer, based on geochemical constraints [2]. Thus, Mn oxide would have formed gradually, and Mn(II) would have dispersed farther from hydrothermal vents than Fe(II) did. The necessary oxic conditions (>2 mL/L dissolved $O_2$) for initiating microbial enzymatic Mn(II) oxidation would have overwhelmed the microbial oxidation of Fe(II) (if it was present).

Mn ores in the Datangpo Formation show the extreme fractionation of Mn with respect to Fe (0.03–0.52; see Supplementary File S2), which can be explained by two possible mechanisms. First, Fe may have been previously fixed as sulfides over the preceding Mesoproterozoic era, causing low sulfate availability (<2 mM) during the Mn metallogenesis period [42,65]. This possibility is evidenced by the very high $\delta^{34}S$ values of pyrites in the basal Datangpo Formation [9,52]. Second, weakly oxygenated conditions locally might have preferentially removed Fe(II), as evidenced by the Sturtian-age BIFs distributed in the underlying sequences of the Datangpo Formation in some areas within the Nanhua Rift Basin [9,60]. In this case, Mn(II) would have remained in an aqueous solution since a strong oxidizing potential is needed for Mn oxide precipitation.

However, for the Otjosondu Mn deposit, the partitioning of Fe and Mn occurred first at exhalative vents due to the preferential precipitation of Fe sulfides and pyritic schist protoliths (Figure 4b). Subsequently, the fractionation of Mn with respect to Fe, i.e., the cyclic deposition of BIFs and Mn layers, was achieved via the shifts in Fe(II)/Fe(III) and Mn(II)/Mn(IV) redoxclines according to the repeated cycles of transgression and regression [3]. The contact between the Mn and Fe layers indicates the alternating activities of Mn- and Fe-oxidizing bacteria controlled by redox conditions in a double-microbial ore-forming system.

### 4.4. Coupling Effect on Mn Metallogenesis

From the above perspective, advantageous conditions contributing to sedimentary Mn metallogenesis were produced due to geo-environmental events during the Sturtian glacial–interglacial transition, as follows:

(1) Oxygen availability. The formation of Mn(IV) oxides required a strong oxidizing potential. The recovery of microorganisms after the Sturtian glaciation promoted marine primary productivity and boosted the $O_2$ levels of atmosphere and a shallow hydrosphere layer.

(2) Redox-stratified restricted basins. The water column underwent a redox change from an overall anoxic state during the Sturtian glaciation to a stratified redox state in its aftermath. The deep anoxic part of a stratified basin served as a storage area for concentrated dissolved Mn(II) that emerged due to hydrothermal activity.

(3) Redox transition from anoxic to oxic. Anoxic water introduced with substantial pre-stored Mn(II) was exchanged for oxygenated water. This process was achieved through transgression–regression cycles and the downwelling of dense surface waters.

We further propose a schematic (Figure 8) to illustrate the major geo-environmental processes and their coupling effects on sedimentary Mn metallogenesis during the Sturtian glacial–interglacial transition. The occurrence of sedimentary Mn deposits during this period is closely linked to glaciogenic sequences. The withdrawal of considerable quantities of seawater during the Sturtian glaciation might have provided specific storage conditions for dissolved Mn(II) and Fe(II) derived mainly from rift-related hydrothermal activity. The evolution of rift basins, as part of the breakup of Rodinia, provided suitable geological settings and sites for Mn accumulation. The microbial biota recovered immediately following the Sturtian glaciation, causing a significant increase in $O_2$ levels in the atmosphere and surface ocean. However, widespread anoxia prevailing in the bottom water allowed it to continue to hold substantial Mn(II) and Fe(II). Exogenic processes, such as transgressive–regressive cycles and the downwelling of oxygenated waters, drove the exchange of redox-stratified water in the rifting basins to facilitate the transfer of the highly concentrated Mn(II) to the shallow drowned shelves in the Otjosondu region, Namibia, as well as to the centers of rift sub-basins in the south-eastern Yangtze Platform, South China.

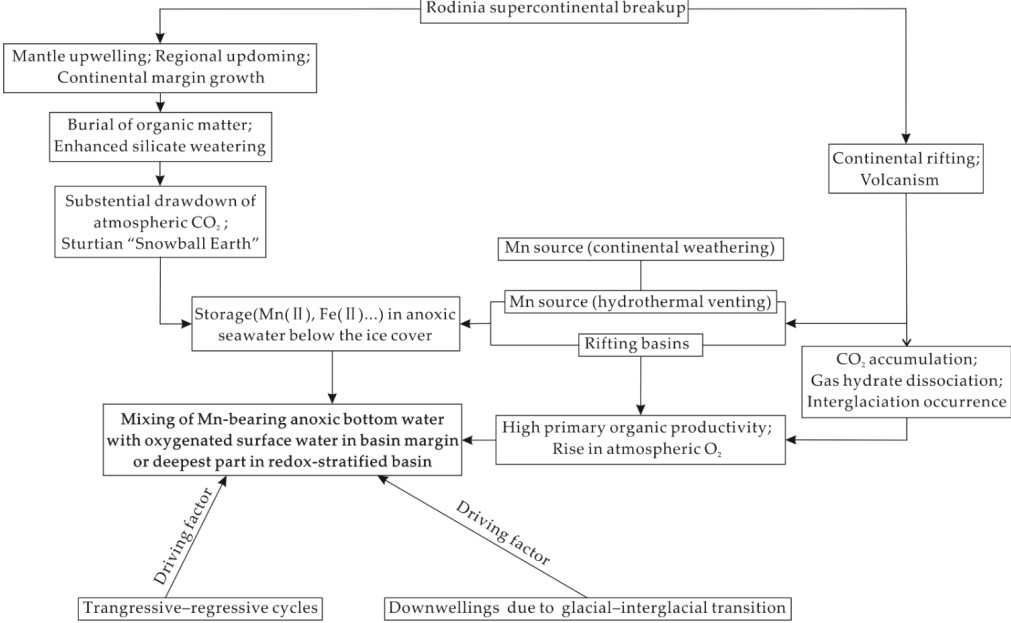

**Figure 8.** Schematic representation showing the relationships among the breakup of Rodinia, climate change, and other geological and environmental events, as well as their coupling with Mn metallogenesis during the Sturtian glacial–interglacial transition.

## 5. Mn Metallogenesis through Geologic Time

Given the geo-environmental scenarios linked to Mn metallogenesis during the Sturtian glacial–interglacial transition, we conducted a brief review of other major sedimentary Mn deposits throughout Earth's history to explore related scenarios of changes in geological and geochemical factors.

### 5.1. Major Sedimentary Mn Deposits throughout Earth's History

Over the course of Earth's history, the water chemistry of marine basins was initially extremely reducing in the early Archean, followed by the evolution of oxygen oases in shallow locales during the late Archean , and finally, by a redox-stratified ocean system in the Paleoproterozoic [66–68]. Thus, in the late Archean, small Mn deposits were found only on basin margins and shallow shelves, where oxygenated oases, limited to small areas, were produced with the evolutionary appearance of Photosystem II (PSII) in the late Archean. PSII, or water-plastoquinone oxidoreductase, uses light to drive oxygenic photosynthesis and is located in the thylakoid membranes of chloroplasts and cyanobacteria [69,70]. However, substantial amounts of Mn(II) and Fe(II) were inferred to have been introduced by the frequent rift-related hydrothermal activity during this period and concentrated in anoxic oceans in their soluble forms [71]. Around the time of the Archean–Proterozoic transition, concomitant with the enhanced burial of organic matter and a decrease in volcanic emissions of reduced gases (e.g., $CH_4$, CO, and $H_2$), a net increase in photosynthetic oxygen production significantly promoted a substantial increase in atmospheric oxygen and triggered the Great Oxidation Event (2.45–2.26 Ga) [2,72]. However, the seawater column was not completely oxidized and remained dominated by redox stratification with oxygenated topwater and anoxic bottom water [5,7]. Regional transgression–regression cycles caused by plate movement or glacial retreat-assisted transfer concentrated Mn(II) in the anoxic parts of stratified basins across an Mn(II)/Mn(IV) redoxcline to precipitate as Mn(IV) oxides on the oxygenated shallow shelf margins. This convergence of factors produced volumetrically large global Mn deposits, including the single largest Mn reserve, the Kalahari Mn field (approximately 2.4 Ga), in the Hotazel Formation of Postmasburg, South Africa [73,74]. Additionally, major black carbonaceous shale–hosted Mn carbonate deposits in the Birimian sequence (ca. 2.3–2.0 Ga), Ghana, West Africa; in the Francevillian sequence (ca. 2.2–2.1 Ga), Gabon, West Africa; in the Sausar Group, central India correlate to this period. Some of these deposits consist of Mn carbonates transformed from Mn oxides during early diagenesis.

The Mesoproterozoic period (ca. 1.6–1.0 Ga) has long been considered an era devoid of Mn deposits, possibly due to its long-term deep-water anoxia. However, sedimentary Mn-rich sequences discovered in China, including the Wafangzi Mn oxide–carbonate deposit (ca. 1.17–1.07 Ga) in Liaoning province [75] and the Qinjiayu Mn carbonate deposit in Hebei province, at least imply the occurrence of episodic oxygen states. Furthermore, these discoveries belie the reputation of the middle Proterozoic as the "boring billion [2]."

The Paleozoic Era was characterized by repeated local glacial cycles, mantle-induced oceanic volcanism, and hydrothermal activities linked to the final breakup of the Rodinian supercontinent and the assembly of Gondwanaland. In these scenarios, Earth's system witnessed the formation of the large Mn oxide–carbonate deposits in Central Kazakhstan (Late Devonian) [76] and several small black-shale-hosted Mn carbonate deposits in Tiantaishan (early Cambrian) [57], Taojiang (Middle Ordovician) [77], and Jiaodingshan (Late Ordovician) [78]. Two major sedimentary Mn carbonate deposits in China were formed in response to oceanic anoxic events and associated glacioeustatic sea level changes: the Xialei deposit in Guangxi province in the Late Devonian and the Zunyi deposit in Guizhou province in the Late Permian.

Anoxic events and sea level fluctuations that may have provided advantageous environments for the source, and the deposition of sedimentary Mn deposits also frequently occurred in the absence of glacial–interglacial cycles during the Mesozoic era [79–81] and early Oligocene epoch [82]. The redox stratification of ocean water in the Mesozoic usually

responded to oceanic anoxic events, which were caused by organic matter biodegradation, $CH_4$ release, or sluggish water circulation in a greenhouse-effected climate [2,83]. The main resulting deposits include major black carbonaceous shale–hosted Mn carbonate deposits in the Jurassic (e.g., Úrkút, Hungary, and Molango, Mexico) [60,64], major Mn oxide and Mn oxide–carbonate deposits in the Cretaceous (e.g., Groote Eylandt, Australia, and Imini, Morocco) [84,85], and colossal Mn oxide–carbonate deposits in the early Oligocene (Chiatura, Georgia, and Nikopol, Ukraine) [2,86]. Since the late Miocene, Mn precipitation has mostly been restricted to the deep ocean floor as Mn nodules and crusts in regions influenced by the activity of the Antarctic Bottom Water, as controlled by interrelated tectonic forces and climatic changes [2,82].

### 5.2. General Models for Mn Metallogenesis

In review of the metallogenesis of major sedimentary Mn deposits in the geological record, the cause of sedimentary Mn deposition has also been strongly linked to the geo-environmental events throughout Earth's history [2,36,70,87]. These events are primarily the breakup of supercontinents and rapid seafloor spreading, and attendant considerations are their consequences for climate change (e.g., glacial–interglacial transition) [5] and endogenic (volcanism and hydrothermal activity) and exogenic evolution (e.g., atmospheric and hydrospheric $O_2$ levels and oceanic anoxic events) [2,88,89]. These processes are capable of producing geological and geochemical conditions suitable for the supply, migration, and precipitation of sedimentary Mn deposits [2]. In general, Precambrian sedimentary Mn deposits were formed in a widespread anoxia ocean with oxic shallow water or episodic deep-water oxygenation, whereas Phanerozoic Mn deposits emerged during episodic anoxic events in an oxic ocean.

A "bathtub ring" model [1,2] has been proposed as a general model suitable for interpreting sedimentary Mn metallogenesis throughout geological time (Figure 9a). In this model, Mn undergoes vertical geochemical cycling, with solid-phase Mn(IV) that precipitates at or just above the redox interface becoming reductively dissolved in deeper anoxic seawater. In this model, solid-phase Mn(IV) particulates are reduced and dissolved unless the substrate (e.g., continental shelves or seafloor) is shallow enough to intercept the redoxcline of a restricted anoxic basin (Figure 9a). Thus, a critical depth for Mn enrichment creates a "bathtub ring." As changes in the relative sea level caused by repeated marine transgression–regression could cause the Mn(II)/Mn(IV) redoxcline to move up and down, various new "rings" are built wherever a new redoxcline impinges upon the basin slope. This model has been used as the basis for several specifically derived models, such as the "oxygen-minimum zone" (Figure 9b) and "episodic ventilation" models (Figure 5b).

All known significant sedimentary Mn enrichment includes primary Mn oxides or Mn carbonates. The formation of Mn carbonate deposits can be explained by primary sedimentary (microbial enzymatic Mn(II) oxidation from the water into the sediment) and diagenetic processes (the redistribution of ore-forming materials via microbe-mediated diagenesis) [86,90]. Although some evidence has demonstrated the direct precipitation of Mn carbonates from a redox-stratified water column (e.g., Fayetteville Green Lake, New York, USA [91]; Brownie Lake, Minnesota, USA [92] and the Orca Basin in the Gulf of Mexico [93]), these occurrences are extremely small scale and cannot be described as significant Mn deposits.

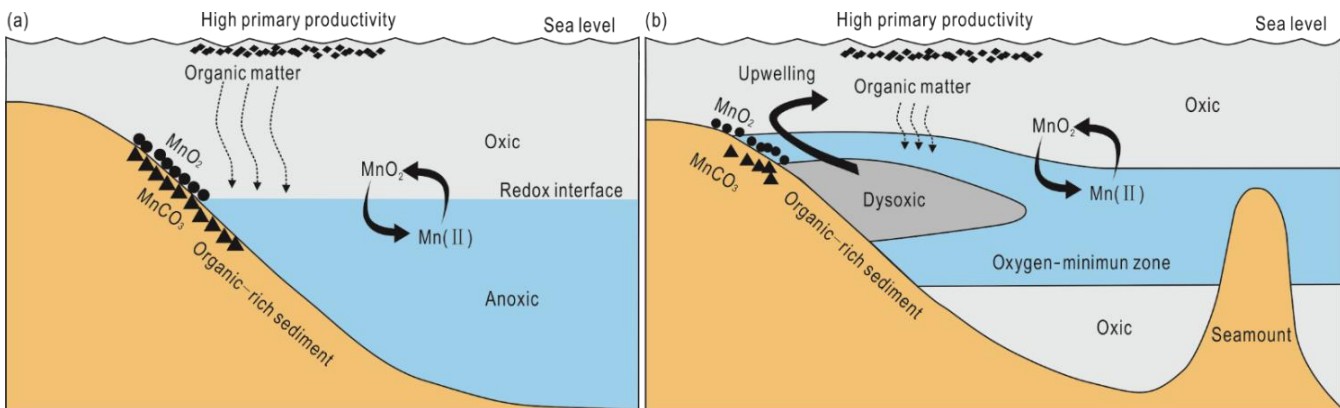

**Figure 9.** (**a**) Bathtub ring model: the traditional model for Mn accumulation in the basin margin (modified from [1,2]); (**b**) oxygen-minimum zone model of Mn mineralization (modified from [94]).

## 6. Conclusions

The Sturtian glacial–interglacial transition period was a critical geological epoch for sedimentary Mn deposition. Mn metallogenesis occurred against a backdrop of dramatic environmental changes in the context of the breakup of Rodinia, the Sturtian glaciation, and the NOE. The interplay of these processes created geologically and geochemically favorable conditions for the supply, accumulation, and precipitation of major Mn ore deposits in Otjosondu and South China.

Rodinia's breakup and the consequent evolution of rift-related basins provided advantageous locations and ore-forming fluids for Mn metallogenesis. During the Sturtian glacial–interglacial transition, the recovery of early microorganisms inhabiting cryoconite holes and crack systems as refugia during the Sturtian glaciation triggered significant increases in primary productivity and atmospheric $O_2$ levels. However, the water system remained redox-stratified, with $O_2$-rich surface water and anoxic deep water.

Exogenic forces, including transgression–regression cycles and $O_2$-rich downwelling, drove the internal exchange of the redox-stratified water mass, precipitating Mn(II) as Mn oxides in rift sub-basin margins (e.g., Otjosondu region) or in rift sub-basin centers (e.g., Datangpo Formation). In the Datangpo Formation, primary Mn oxides were then diagenetically reduced to Mn carbonates via the oxidation of organic matter. Microbial activity plays a cooperative role in Mn primary deposition and subsequent diagenesis.

Furthermore, a brief review of Mn metallogenesis in the geological record, including the Mn deposits formed in the context of the Sturtian glacial–interglacial transition, revealed that the geological and geochemical conditions necessary for Mn metallogenesis were achieved in different ways at different times during Earth's history. However, all cases involved the interplay of geologic forcings and endogenic and exogenic environmental processes.

**Supplementary Materials:** The following supporting information can be downloaded at: https://www.mdpi.com/article/10.3390/min13060712/s1, Supplementary File S1. The locations of Mn deposits [23,63]; Supplementary File S2. Original data and references [23,95–106].

**Author Contributions:** L.L. wrote the paper; Z.J. contributed to information collection. F.C. supervised and provided critical reviews. All authors have read and agreed to the published version of the manuscript.

**Funding:** The authors gratefully acknowledge the financial support from the Scientific Research Fund of the Second Institute of Oceanography, MNR, grand No. JB2303 and the 13th Five–Year Plan of the China Ocean Mineral Resource R&D Association (COMRA) (Grant Nos. DY135-N1-1, DY135-N2-1, and DY135-N1-1-01).

**Data Availability Statement:** The authors confirm that the data supporting the findings of this study are available within the Supplementary Materials.

**Acknowledgments:** We would like to thank the reviewers for constructive feedback and valuable suggestions and editors for editing the article.

**Conflicts of Interest:** The authors declare no conflict of interest.

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
