# Peer review of "Sedimentary Mn Metallogenesis and Coupling among Major Geo-Environmental Events during the Sturtian Glacial–Interglacial Transition"

_minerals, doi:10.3390/min13060712_

Round 1
Reviewer 1 Report
Liu et al. reviewed the current understanding on the mechanisms of Mn metallogenesis during the Sturtian glacial–interglacial transition by comparing the Otjosondu area and the Datangpo Formation. I have a few major comments that I hope the authors can address before publication.
Major comments
1. It is not clear to me whether the authors included any new data in the paper. In the introduction the paper does say “We also collected a series of major and rare earth elements and stable carbon and oxygen isotopes of some Mn ore and host rock samples from the Datangpo Formation and some selected Mn deposits in the geological record.” Similar statements also appear on page 7: “We have collected many major rare earth elements (REEs + Y, REY) of Mn carbonate ores and host rocks”. However, I don’t see any sample location, sample processing methodology and results. The supplementary data table only includes previously published data. In the supplementary figure, I also don’t see where the sample locations are. If the authors meant to just compile the published data, they should say it clearly by modifying this statement. Otherwise it’s very confusing where the data goes.
2. The authors need to elaborate more on how ‘the southward drift of the large Laurentian magmatic province [17] mitigated CO2 consumption by continental weathering’. Because the magmatic province migrated out of tropics…? Or some other reasons?
3. Please check Figure 3 as I found multiple inconsistencies with the original publications. For instance, FeHR/FeT>0.8 and 0.6-0.8 in the 1st number is categorized as oxic-suboxic, even though the Fe speciation proxy treats FeHR/FeT>0.38 as anoxic. If the formation depth is the same as Xu(2019), then the Cr isotope mean values for different intervals are not correct (reversed compared to the original publication). The Cr isotope data on the rhodochrosite ore should average at 0.01 ± 0.10‰, whereas the shale has a mean value of -0.10 ± 0.06‰. Interpretation based on Fe isotopes is also not accurate for the top section in the 2nd member. The original publication (Zhang et al., 2015) refer to the 2nd member as “near-quantitative oxidation of ferrous iron to Fe-oxyhydroxides followed by near-quantitative reduction and conversion to pyrite in the local diagenetic environment”. Given this, the 2ndmember should be oxic-suboxic water columns that allow Fe oxide formation, and only the porewater is sulfidic to allow post-depositional pyrite formation.
4. It is not clear how “oceanic upwelling changes during operations of transgressions and regressions” in section 4.1.1. (upper page 6). In the modern ocean, upwelling mainly occurred due to wind forcing (e.g., equator, southern westerlies, and coastal upwelling due to the Ekman transport by the eastern boundary currents like Peru and California margin). How transgression/regression change ocean upwelling needs to be explained. Also as the supercontinent broke up, chances are that upwelling zones were shifted with changing wind fields dramatically. The authors should have a more thorough description of the paleogeography of the two sites mentioned in the manuscript and elaborate on how upwelling could have been affected transgression/regression cycles.
5. What caused downwelling? Did the dense water come from ice sheet melting during deglaciation? This process needs to be explained better.
Minor comments
1. Abstract: “tortuous” – do you mean ‘sustained’ or ‘prolonged’? ‘The water system (column) was not completely oxidized and (has) sustained redox stratification.
2. Introduction paragraph 1: ‘low oxic’ should be ‘anoxic’. Under high oxic conditions and high pH (>9.5), insoluble oxidized Mn species (Mn3+(III), Mn4+(IV)) are thermodynamically favored’. In seawater that’s close to neutral pH, Mn(III)/Mn(IV) oxides are also favorable, so no need for pH to be >9.5.
3. Section 2: ‘Many chronological studies, including Re–Os isotope and U–Pb SHRIMP zircon dating’. Define ‘SHRIMP’.
4. Section 3.1, page 4: ‘Organic matter is buried, and O2 is added to the atmosphere if anaerobic respiration is incomplete.’ This is not clear. What do you mean by incomplete anaerobic respiration? There is always TOC to be respired. Oxygen is added to the atmosphere no matter anaerobic respiration occurs or not since it’s only relevant with photosynthesis.
5. Section 4.1.1., page 6: The cyclic depositional rhythm of BIFs and Mn layers, which resembled sandwiches, had been depended (dependent) on the shifting of Fe2+/Fe3+ and Mn2+/Mn4+ redoxclines as a result of repeated transgression–regression cycles.
6. Section 4.1.2, page 7: “Other redox proxies (U, Mo, V, et al.) analyzed from the entire profile of Datangpo Formation [46] suggested that the depositional environment for Mn layers was oxic, whereas that for the host rocks was anoxic”. This is not clear. Do you mean layers with Mn enrichments were deposited in oxic environments and but the other layers are not? Or do you mean the water column was oxic to allow Mn accumulation but the porewater was anoxic to allow diagenesis of Mn oxides to Mn carbonate?
7. Section 4.2: “Second, weakly oxygenated conditions at locales might have preferentially removed Fe2+”.
8. Section 4.4, page 10: Define ‘Photosystem II’. What is ‘reduced gas’ in this sentence: ‘concomitant with the enhanced burial of organic matter and reduction in reduced gases’?
9. Page 11: ‘Regional transgression–regression cycles caused by plate movement or glacial retreat-assisted transfer concentrated Mn2+ in the anoxic part of the stratified basin across Mn2+/Mn4+ redoxcline to precipitate as Mn4+(III)/(IV) oxides on the oxygenated shallow shelf margins.
10. Page 12: ‘This process can be driven by transgression–regression cycles, upwellings, and dense downwellings.’
11. Page 12: “Solid-phase Mn4+(III)/(IV) particulates would have been reduced unless the substrate (e.g., continental shelves or seafloor) is shallow enough to intercept the redoxcline of a restricted anoxic basin”
12. Why are data not available? This is not what current journals would be up for, unless the data are confidential due to some reason.
13. Acknowledgements are missing.
Author Response
Reply to the comments of Reviewer #1
We sincerely thank you for your thorough review and valuable suggestions. Your feedback is crucial for enhancing our manuscript.
We have revised the manuscript for language and grammar with the assistance of a native English-speaking editor.
Major comments
- It is not clear to me whether the authors included any new data in the paper. In the introduction the paper does say “We also collected a series of major and rare earth elements and stable carbon and oxygen isotopes of some Mn ore and host rock samples from the Datangpo Formation and some selected Mn deposits in the geological record.” Similar statements also appear on page 7: “We have collected many major rare earth elements (REEs + Y, REY) of Mn carbonate ores and host rocks”. However, I don’t see any sample location, sample processing methodology and results. The supplementary data table only includes previously published data. In the supplementary figure, I also don’t see where the sample locations are. If the authors meant to just compile the published data, they should say it clearly by modifying this statement. Otherwise it’s very confusing where the data goes.
Author:
Thank you for your comments.
Yes, the supplementary data is derived from previously published literature. We have modified the statement in the Introduction on Page 2.: “We also compile a series of published data, including major and rare-earth elements and stable carbon and oxygen isotopes of Mn ore and host rock samples from the Datangpo Formation and selected Mn deposits throughout the geological record.”
Furthermore, we have included a figure showing sample locations of the Datangpo Mn deposits in the Supplementary File 1.
- The authors need to elaborate more on how ‘the southward drift of the large Laurentian magmatic province [17] mitigated CO2 consumption by continental weathering’. Because the magmatic province migrated out of tropics…? Or some other reasons?
Author:
Thank you for pointing this out. We have elaborated more on this point in paragraph 2 of Section 2 (Page 2).
- Please check Figure 3 as I found multiple inconsistencies with the original publications. For instance,
1) FeHR/FeT>0.8 and 0.6-0.8 in the 1st number is categorized as oxic-suboxic, even though the Fe speciation proxy treats FeHR/FeT>0.38 as anoxic.
2) If the formation depth is the same as Xu(2019), then the Cr isotope mean values for different intervals are not correct (reversed compared to the original publication). The Cr isotope data on the rhodochrosite ore should average at 0.01 ± 0.10‰, whereas the shale has a mean value of -0.10 ± 0.06‰.
3) Interpretation based on Fe isotopes is also not accurate for the top section in the 2nd member. The original publication (Zhang et al., 2015) refer to the 2nd member as “near-quantitative oxidation of ferrous iron to Fe-oxyhydroxides followed by near-quantitative reduction and conversion to pyrite in the local diagenetic environment”. Given this, the 2ndmember should be oxic-suboxic water columns that allow Fe oxide formation, and only the porewater is sulfidic to allow post-depositional pyrite formation.
Author:
We are grateful for your careful review of Figure 3. We have rechecked these original articles and made the following revisions.
1) You are correct. After reviewing the two original articles, we found that the authors did not state whether the samples they collected were Mn ores or host rocks of the basal Datangpo Formation. Thus, the data did not represent the depositional conditions of the Mn layers.
We have now cited a recently published paper ([46], Ai et al., 2021) that reports Fe species data for the Mn layers and host rocks of the 1st and 2nd members of the Datangpo Formation.
2) Yes. We have made the necessary revisions.
3) The “2nd interval” in the original paper does not represent the 2nd member of the Datangpo Formation. It belongs to the basal (1st member) of the Datangpo Formation. Please refer to Figure 3 in the original paper ([48], Zhang et al., 2015).
- It is not clear how “oceanic upwelling changes during operations of transgressions and regressions” in section 4.1.1. (upper page 6). In the modern ocean, upwelling mainly occurred due to wind forcing (e.g., equator, southern westerlies, and coastal upwelling due to the Ekman transport by the eastern boundary currents like Peru and California margin). How transgression/regression change ocean upwelling needs to be explained. Also as the supercontinent broke up, chances are that upwelling zones were shifted with changing wind fields dramatically. The authors should have a more thorough description of the paleogeography of the two sites mentioned in the manuscript and elaborate on how upwelling could have been affected transgression/regression cycles.
Author:
Thank you for your valuable input. Unfortunately, we were unable to find any materials on the paleogeographical setting materials that could elaborate on how upwelling was affected by transgression–regression cycles.
We have thoroughly reviewed the literature regarding the ancient sedimentary Mn deposits throughout geological time. Although researchers have proposed general models of Mn metallogenesis based on the transgression–regression cycle, they have yet to further elaborate on how upwellings occurred. This may be due in part to subsequent plate movements disrupting geological settings of metallogenetic Era and partially because authors have been unable to observe the in situ geo-environmental factors of the geological time in the present day.
- What caused downwelling? Did the dense water come from ice sheet melting during deglaciation? This process needs to be explained better.
Author:
Thank you for your suggestion. We have provided further elaboration on this point in paragraph 1 of on Page 8.
Minor comments
- Abstract: “tortuous” – do you mean ‘sustained’ or ‘prolonged’? ‘The water system (column) was not completely oxidized and (has) sustained redox stratification.
Author:
1) The term “tortuous” implies that “Sea ice might have remained in the aftermath of Sturtian glaciation, and complete deglaciation might not have been as abrupt as previously thought.” To avoid ambiguity, we replaced this word in the abstract with “during the Sturtian glacial–interglacial transition.” The description of “tortuous transition” can be found in paragraph 1 of Page 8.
2) We have revised the manuscript for language and grammar with the assistance of a native English-speaking editor. Thank you for carefully identifying these grammar errors.
- Introduction paragraph 1: ‘low oxic’ should be ‘anoxic’. Under high oxic conditions and high pH (>9.5), insoluble oxidized Mn species (Mn3+(III), Mn4+(IV)) are thermodynamically favored’. In seawater that’s close to neutral pH, Mn(III)/Mn(IV) oxides are also favorable, so no need for pH to be >9.5.
Author:
We appreciate your suggestion and have made the necessary revisions in paragraph 1 of the Introduction section.
- Section 2: ‘Many chronological studies, including Re–Os isotope and U–Pb SHRIMP zircon dating’. Define ‘SHRIMP’.
Author:
Thank you. We have defined “SHRIMP” in the last paragraph on Page 2.
- Section 3.1, page 4: ‘Organic matter is buried, and O2 is added to the atmosphere if anaerobic respiration is incomplete.’ This is not clear. What do you mean by incomplete anaerobic respiration? There is always TOC to be respired. Oxygen is added to the atmosphere no matter anaerobic respiration occurs or not since it’s only relevant with photosynthesis.
Author:
Thank you. After reviewing the reference cited here, we agree with your input and have deleted “if anaerobic respiration is incomplete” from this sentence (on Page 4 after reorganizing the manuscript).
- Section 4.1.1., page 6: The cyclic depositional rhythm of BIFs and Mn layers, which resembled sandwiches, had been depended (dependent) on the shifting of Fe2+/Fe3+ and Mn2+/Mn4+ redoxclines as a result of repeated transgression–regression cycles.
Author:
Thank you. We have revised this grammar error.
- Section 4.1.2, page 7: “Other redox proxies (U, Mo, V, et al.) analyzed from the entire profile of Datangpo Formation [46] suggested that the depositional environment for Mn layers was oxic, whereas that for the host rocks was anoxic”. This is not clear. Do you mean layers with Mn enrichments were deposited in oxic environments and but the other layers are not? Or do you mean the water column was oxic to allow Mn accumulation but the porewater was anoxic to allow diagenesis of Mn oxides to Mn carbonate?
Author:
Thank you for pointing out this unclear statement.
The Mn-bearing sequences are hosted in the basal Datangpo Formation (see Figure 5a) and interbedded with black carbonaceous shales in a series of graben sub-basins. The interbedded black carbonaceous shales serve as the host rocks. The redox proxies indicate that the depositional environment for Mn layers was oxidative; however, the host rocks were deposited under anoxic water conditions.
We have provided further elaboration on this point on Pages 6 and 7.
- Section 4.2: “Second, weaklyoxygenated conditions at locales might have preferentially removed Fe2+”.
Author:
Thank you for your careful review. We have revised these language errors.
- Section 4.4, page 10: Define ‘Photosystem II’. What is ‘reduced gas’ in this sentence: ‘concomitant with the enhanced burial of organic matter and reduction in reduced gases’?
Author:
Thank you.
1) We have defined “Photosystem II” on Page 11 (after reorganizing the manuscript).
2) The term “reduced gases” has been used in many articles, such as the following articles.
[1]. Kasting, J. Earth’s early atmosphere. Science, 1993, 259: 920-926.
[2]. Sleep, N.H. Oxygenating the atmosphere. Nature, 2001, 410: 317-318.
[3]. Holland, H.D. Oxygenation of the atmosphere and oceans. Philos Trans R Soc Lond B Biol, 2006, 361: 903-915.
For a clearer understanding, we have replaced “reduced gases” with “volcanic emissions of reduced gases (e.g., CH4, CO, and H2)” in the last paragraph on Page 11 (after reorganizing the manuscript).
- Page 11: ‘Regional transgression–regression cycles caused by plate movement or glacial retreat-assisted transfer concentrated Mn2+ in the anoxic part of the stratified basin across Mn2+/Mn4+ redoxcline to precipitate as Mn4+(III)/(IV) oxides on the oxygenated shallow shelf margins.
Author:
We appreciate your suggestion. We have replaced “Mn2+ and Mn4+” with “Mn(II) and Mn(IV)” in the manuscript.
- Page 12: ‘This process can be driven by transgression–regression cycles, upwellings, and dense downwellings.’
Author:
We have made the necessary revisions: “This process was achieved through transgression–regression cycles and the downwelling of dense surface waters.” (on Page 10 after reorganizing the manuscript)
- Page 12: “Solid-phase Mn4+(III)/(IV) particulates would have been reduced unless the substrate (e.g., continental shelves or seafloor) is shallow enough to intercept the redoxcline of a restricted anoxic basin”
Author:
We appreciate your input. We have replaced “Mn2+ and Mn4+” with “Mn(II) and Mn(IV)” in the manuscript.
- Why are data not available? This is not what current journals would be up for, unless the data are confidential due to some reason.
Author:
Thank you for pointing this out. The data are not confidential.
The authors confirm that the data supporting the findings of this study are available within the supplementary materials.
- Acknowledgements are missing.
Author:
Thank you for your suggestion. We have included the “Acknowledgments” in the final version of the manuscript.

Reviewer 2 Report
REVIEW
The manuscript is up to date, fits well with the policy of journal Minerals, well organized and well written. Reviewer agree with the main points of Mn deposit formation via geological ages, concerning the metal source as rifting, hydrothermal discharge system and the fundamental role of oxygen supply (need of obligatory oxic conditions for Mn and suboxic conditions for Fe, which can be the result of different processes, like transgression/regression, ventilation, etc.).
Sorry to say, that the authors forgot to mention the most important, fundamental fact for the Mn deposit formation, and the enrichment process, the microbial Mn oxidation. This results the Mn oxide minerals in the sediment and also remarkable mass of organic matter.
In low T aquatic systems (T<150 °C, sedimentary environment) manganese precipitation is microbial, and first sequestration of the dissolved Mn(II) needs obligatory oxic conditions, because the multicopper oxidaze enzymatic process active only under the obligatory oxic conditions. Mn(II) precipitation is microbial, which have atomic structure (electron configuration) reasons, because the transmission of the two electrons to form Mn4+ have different energy content. The third electron transmission: (Mn3+/Mn2+): dGo: 67 kJ/mol (energy demand), the fourth electron transmission: (MnO2/ Mn3+): dGo: -137 kJ/mol. (Bargar et al., 2004, 2005; Webb et al,. 2005; Morgan, 2005).
So, not an abiogenic redox system, but redox is the demand of enzymatic activity which is the main factor of metal enrichment. The Mn and Fe deposits are so called redox deposits, but we ask for attention that this fact is not in abiogenic meaning. The enzymatic activity has redox demand, which is obligatory oxic in the case of Mn oxidizing bacteria and suboxic in the case of Fe oxidizing bacteria. These deposits are microbialite ores, with 2 step microbially mediated formation model.
A. Syngenetic formation – source of elements – is similar
- Geodynamic situation refers to rifting (or failed rifting) zone, and distal hydrothermal discharge as metal source
- Oxygen supply is basically important, suboxic in the case of Fe and obligatory oxic in the case of Mn (to form a deposit, a very effective enzymatic enrichment engine is needed). The oxygen supply can be the result of currents, cyanobacterial ventilation – and its nuance change can determine, what kind of deposit will be formed, or not formed (suboxic – Fe, obligatory oxic – Mn)
- Further needed condition: starving basins. If there is a considerable debris contribution, microbial biomat system will be destroyed
- Enrichment process – enzymatic selective element enrichment – very effective (enzymatic engine and its redox demand)
B. Diagenesis (Two step microbially mediated formation model).
Taking into consideration, the manuscript needs complement based on the above aspects. For details see Minerals 2022, 12(10), 1273. (https://doi.org/10.3390/min12101273.
Reviewer
Author Response
Reply to the comments of Reviewer #2:
Thank you for your constructive feedback and valuable suggestions. We have thoroughly read the paper by Polgári and Gyollai (2022), which you kindly shared with us. Additionally, we have read the articles listed in the reference section, particularly [1-12]. These papers provided us with a deeper understanding of the microbial activities in the role of Mn metallogenesis, allowing us to make complementary additions to sections 4.2 and 4.3.
“4.2 Contribution of microbial processes to Mn metallogenesis
In addition to the geochemical behavior of Mn, microbially mediated Mn fixation plays an important role in considerable Mn accumulation in sediments. Mn deposits across different geological periods exhibit a similar two-step microbial metallogenesis (Polgári et al., 2012, 2019). 1) Primary chemolithoautotrophic cycle: microbes oxidize Mn(II) to Mn(IV) via an enzymatic pathway under essential oxic conditions. 2) Diagenetic heterotrophic cycle: this cycle involves the decomposition and mineralization of cells and extracellular polymeric substances (EPSs) of Mn (and Fe) bacteria.
In the case of Mn carbonate deposits within black shale, microbes mediate Mn-oxide reduction via organic matter decomposition, resulting in the formation of rhodochrosite, characterized by a light δ13Ccarb signal, as the final product (Yu et al., 2019; Polgári and Gyollai, 2022). However, for Mn-oxide deposits, such as those found in the Otjosondy region of Namibia, Mn oxides do not transform into Mn carbonates, either due to a slow organic matter sedimentation rate or because the majority of the organic matter was oxidized under oxic conditions throughout the process of metallogenesis.
Although diagenesis and other processes can obscure microbial characteristics, remnants of microbial activity, including mineralized microbial microtextures and fossils of various shapes (Polgári et al., 2012, 2019; Yu et al., 2019), remain preserved, likely owing to the protective nature of the minerals around the cells and EPS. Polgári and Gyollai (2022) suggest a multiscale methodology to verify the role of microbe-mediated ore-forming processes.”
The following sentences have been added to “Section 4.3 Mn and Fe fractionation”:
“The necessary oxic conditions (>2 mL/L dissolved O2) for initiating microbial enzymatic Mn(II) oxidation would have overwhelmed the microbial oxidation of Fe(II) (if present).
The contact between the Mn and Fe layers indicates the alternating activities of Mn- and Fe-oxidizing bacteria controlled by redox conditions in a double microbial ore forming system.”
Moreover, we have revised the manuscript for language and grammar with the assistance of a native English-speaking editor.
Thank you once again for your feedback.

Reviewer 3 Report
This paper privided a brief review on some aspects of the metallogenesis of Mn ore deposits. Basically confrom to the general idea of the Mn metallogenesis. A better organization is still required, especially section 4.4, it is too long, and did not follow a appropriate organization. In section 3, reorganization is needed, authors should list out specifically what kind of geological events occurred during that stage. The author should also understand the different between geological events and the environmental condition of the basin. Is "redox stratified water system" a geological events ? I don't think so. I understand that the paper was translated directly from Chinese, but please organize the paper before the translation. The direct link between the geological events and the metallogenesis is also not clear. In general, the idea of the paper is OK, but a lot revision is still needed, better organisation and language is needed. The author need to be clear, what kind of gelogical events is associated with the formation of Mn ore. List them out in section 3 one by one. Then discuss the correlation between them one by one in section for, provide evidence accordingly. Organize the material wisely.
The general idea of this paper is good, because indeed, the development of the Mn ore deposits correspond to the geological events. It is defininitely a paper worth publishing. I recommand accept after necessary revision is made.
1. “The main points are as follows.” "Against this background"."
In the context of the Rodinia breakup and Sturtian glacial–interglacial transition, dra- matic exogenic environmental changes were experienced globally." "oases or oasis"Please try not to translate Chinese directly into English, and use the apprropriate punctuation. Moderate English changes is required before it can be published. Most of the expression is translate from Chinese directly.
The highlights in the pdf file mark the errors of the languages that need to be revised. But be aware that there are a lot of grammar and phrasing and wording mistakes that don't limit to the marked words, and need to be attended to thoroughly.

Author Response
Reply to the comments of Reviewer #3
This paper privided a brief review on some aspects of the metallogenesis of Mn ore deposits. Basically confrom to the general idea of the Mn metallogenesis. A better organization is still required, especially section 4.4, it is too long, and did not follow a appropriate organization. In section 3, reorganization is needed, authors should list out specifically what kind of geological events occurred during that stage. The author should also understand the different between geological events and the environmental condition of the basin. Is "redox stratified water system" a geological events ? I don't think so. I understand that the paper was translated directly from Chinese, but please organize the paper before the translation. The direct link between the geological events and the metallogenesis is also not clear. In general, the idea of the paper is OK, but a lot revision is still needed, better organisation and language is needed. The author need to be clear, what kind of gelogical events is associated with the formation of Mn ore. List them out in section 3 one by one. Then discuss the correlation between them one by one in section for, provide evidence accordingly. Organize the material wisely.
The general idea of this paper is good, because indeed, the development of the Mn ore deposits correspond to the geological events. It is defininitely a paper worth publishing. I recommand accept after necessary revision is made.
Author:
Thank you for your constructive feedback and valuable suggestions. We must say that your feedback is indeed crucial for improving our paper.
Firstly, we have replaced “geological events” with “geo-environmental events” in the title and the subtitle of Section 3.
Secondly, we have reorganized the manuscript. We present a brief overview of the contents of the manuscript below.

Section 3 discusses the geo-environmental events associated with Mn metallogenesis during the Sturtian glacial–interglacial transition in Sections 3.1, 3.2, and 3.3, respectively. The favorable conditions for Mn metallogenesis, produced by these events and their interactions, are discussed in Section 4.4, after introducing the two Mn deposits in Section 4.1.
- “The main points are as follows.” "Against this background"."
In the context of the Rodinia breakup and Sturtian glacial–interglacial transition, dra- matic exogenic environmental changes were experienced globally." "oases or oasis"Please try not to translate Chinese directly into English, and use the apprropriate punctuation. Moderate English changes is required before it can be published. Most of the expression is translate from Chinese directly.
The highlights in the pdf file mark the errors of the languages that need to be revised. But be aware that there are a lot of grammar and phrasing and wording mistakes that don't limit to the marked words, and need to be attended to thoroughly.
Author:
Thank you very much for your thorough review. We have rechecked the errors you kindly pointed out. Additionally, we have made revisions to the manuscript in terms of language and grammar with the assistance of a native English-speaking editor.
Thank you once again for your feedback.

Round 2
Reviewer 1 Report
Lines 252-254: Meltwater from the glacier is usually much fresher than the seawater, and thus should have lower density than salty seawater and wouldn't sink to the bottom. Surface water freshening actually may disrupt any deep water formation. Given that the deglaciation might not be abrupt, I would think additional sea ice formation (which makes surface water denser due to brine rejection) would contribute to replenishing anoxic bottom water.
Author Response
Reply to the comments of Reviewer #1 (Round 2)
Lines 252-254: Meltwater from the glacier is usually much fresher than the seawater, and thus should have lower density than salty seawater and wouldn't sink to the bottom. Surface water freshening actually may disrupt any deep water formation. Given that the deglaciation might not be abrupt, I would think additional sea ice formation (which makes surface water denser due to brine rejection) would contribute to replenishing anoxic bottom water.
Author:
Thank for your careful review.
We appreciate your suggestion and have made necessary revisions in paragraph 1 on Page 9.
Thank you once again for your feedback.

Reviewer 2 Report
I accept the inserted parts concerning microbial contribution, so I suggest the manuscript be accepted.
Reading the Ms I found some cases, where the geographical names and the authors' names were not correct, some spelling errors occurred, so please check it.
Author Response
I accept the inserted parts concerning microbial contribution, so I suggest the manuscript be accepted.
Reading the Ms I found some cases, where the geographical names and the authors' names were not correct, some spelling errors occurred, so please check it.
Author:
We are grateful for your careful review of the manuscript.
We have rechecked the spelling errors and made the necessary revisions on Pages 3, 7, 10, 13, 14, 16 and the “References” section. The detailed revisions have been marked up in the manuscript.
Thank you once again for your feedback.
